# Conserved NIMA kinases regulate multiple steps of endocytic trafficking

**Braveen B. Joseph**[1], **Naava Naslavsky**[2,3], **Shaonil Binti**[1], **Sylvia Conquest**[1], **Lexi Robison**[1], **Ge Bai**[4], **Rafael O. Homer**[1], **Barth D. Grant**[4], **Steve Caplan**[2,3], **David S. Fay**[1]*

1 Department of Molecular Biology, College of Agriculture Life Sciences, and Natural Resources, University of Wyoming, Laramie, Wyoming, United States of America, 2 Department of Biochemistry & Molecular Biology, University of Nebraska Medical Center, Omaha, Nebraska, United States of America, 3 The Fred and Pamela Buffett Cancer Center, University of Nebraska Medical Center, Omaha, Nebraska, United States of America, 4 Department of Molecular Biology and Biochemistry, Rutgers University, Piscataway, New Jersey, United States of America

* davidfay@uwyo.edu

**Data Availability Statement:** The authors confirm that all data underlying the findings are fully available without restriction. All relevant data are within the paper and its Supporting Information files. Raw data contained in S1 File.

## Abstract

Human NIMA-related kinases have primarily been studied for their roles in cell cycle progression (NEK1/2/6/7/9), checkpoint–DNA-damage control (NEK1/2/4/5/10/11), and ciliogenesis (NEK1/4/8). We previously showed that *Caenorhabditis elegans* NEKL-2 (NEK8/9 homolog) and NEKL-3 (NEK6/7 homolog) regulate apical clathrin-mediated endocytosis (CME) in the worm epidermis and are essential for molting. Here we show that NEKL-2 and NEKL-3 also have distinct roles in controlling endosome function and morphology. Specifically, loss of NEKL-2 led to enlarged early endosomes with long tubular extensions but showed minimal effects on other compartments. In contrast, NEKL-3 depletion caused pronounced defects in early, late, and recycling endosomes. Consistently, NEKL-2 was strongly localized to early endosomes, whereas NEKL-3 was localized to multiple endosomal compartments. Loss of NEKLs also led to variable defects in the recycling of two resident cargoes of the trans-Golgi network (TGN), MIG-14/Wntless and TGN-38/TGN38, which were missorted to lysosomes after NEKL depletion. In addition, defects were observed in the uptake of clathrin-dependent (SMA-6/Type I BMP receptor) and independent cargoes (DAF-4/Type II BMP receptor) from the basolateral surface of epidermal cells after NEKL-2 or NEKL-3 depletion. Complementary studies in human cell lines further showed that siRNA knockdown of the NEKL-3 orthologs NEK6 and NEK7 led to missorting of the mannose 6-phosphate receptor from endosomes. Moreover, in multiple human cell types, depletion of NEK6 or NEK7 disrupted both early and recycling endosomal compartments, including the presence of excess tubulation within recycling endosomes, a defect also observed after NEKL-3 depletion in worms. Thus, NIMA family kinases carry out multiple functions during endocytosis in both worms and humans, consistent with our previous observation that human NEKL-3 orthologs can rescue molting and trafficking defects in *C. elegans nekl-3* mutants. Our findings suggest that trafficking defects could underlie some of the proposed roles for NEK kinases in human disease.

**Funding:** This project was supported by NIH R35
GM136236 to DSF, BBJ, SB, SC, LR and ROH
(University of Wyoming), NIH R01 GM135326 to
BDG, GB (Rutgers, State University of New Jersey),
NIH R35 GM144102 to SC and NN (University of
Nebraska Medical Center), and by an Institutional
Development Award (IDeA) from the National
Institute of General Medical Sciences of the
National Institutes of Health (P20GM103432) to
DF, BBJ and SB (University of Wyoming). The
funders had no role in study design, data collection
and analysis, decision to publish, or preparation of
the manuscript.

**Competing interests:** The authors have declared
that no competing interests exist.

## Author summary

Intracellular trafficking is an evolutionary conserved process whereby cargoes, which
include proteins, lipids, and other macromolecules, are internalized by cells, packaged
into vesicles, and distributed to their proper places within the cell. This study demon-
strated that two conserved NIMA-related kinases, NEKL-2 and NEKL-3, are required for
the transport of multiple cargoes in the epidermis of *C. elegans*. NEKL-2 and NEKL-3
function at organelles, called endosomes, to regulate their morphology and control the
sorting of cargoes between different intracellular compartments. In the absence of NEKL
activities, various cargoes, including components of the BMP and Wnt signaling path-
ways, were misregulated. Our studies are further supported by results showing that the
human counterparts of NEKL-3, NEK6 and NEK7, were also required for maintaining
endosome morphologies and for the proper sorting of cargo in human cells. Notably,
NIMA-kinases are well studied for their roles in cell cycle regulation, and overexpression
of these kinases is linked to cancer formation and poor prognosis. Our study suggests
their role in cancer progression could be partly due to the abnormal intracellular traffick-
ing of conserved signaling components with known roles in cancer formation.

## Introduction

The NIMA (never in mitosis gene a) family of serine/threonine kinases was first identified in a
genetic screen for mutants that fail to enter mitosis in the filamentous fungus *Aspergillus nidu-
lans* [1–4]. The human genome encodes 11 NIMA-related kinases or NEKs (NEK1–NEK11).
Consistent with the pro-mitotic function of NIMA in *A. nidulans*, several mammalian NEKs
regulate cell division processes including spindle assembly and centrosome separation (NEK1,
NEK2, NEK5–NEK7, and NEK9) and cytokinesis (NEK1, NEK2, NEK6, and NEK7) [5–16].
Correspondingly, misregulation of NEKs can cause aberrant cell proliferation, and overexpres-
sion of several NEKs, including NEK6, NEK7, and NEK9, is associated with multiple cancers
and cardiac hypertrophy [8,17–20]. More recently, attention on NEK7 has focused on its role
as an activator of NLRP3-inflammasomes, which are critical for the immune response against
microbial pathogens [21,22]. Other NEK family members (e.g., NEK1, NEK4, and NEK8)
were identified as causal factors in mouse models of polycystic kidney disease or have been
linked to human ciliopathies [23–30]. Several NEK family members, including NEK8 and
NEK9, span multiple functional categories, as NEK8 has been linked to cancer and cell cycle
control [18,26,31,32] and NEK9 has been implicated in ciliogenesis and ciliopathies [25,33].
These findings suggest that individual NEKs have tissue- or cell type–specific functions, which
may affect different developmental and disease-associated processes.

*C. elegans* has four NIMA-related kinases, NEKL-1, NEKL-2, NEKL-3, and NEKL-4.
NEKL-2, a homolog of human NEK8 and NEK9, contains a highly conserved kinase domain
and a short disordered region at the C terminus [34]. NEKL-2 is expressed in a punctate pat-
tern in the large epidermal syncytium, hyp7, where it colocalizes with the conserved ankyrin
repeat proteins MLT-2 (ANKS6 homolog) and MLT-4 (INVS homolog). MLT-2 and MLT-4
are required for the proper localization of NEKL-2, and their physical association is conserved
in vertebrates [34–36]. NEKL-3, a close homolog of human NEK6 and NEK7, contains a
kinase domain only and is also expressed in a punctate pattern within hyp7. NEKL-3 forms a
complex with a third conserved ankyrin repeat protein, MLT-3/ANKS3, which is required for
the proper localization of NEKL-3 in *C. elegans* and NEK7 in vertebrates [37]. We have shown

that NEKL-2 and NEKL-3, along with their MLT binding partners, are required for the completion of molting in *C. elegans*, as a strong loss of function in any of these proteins leads to molting defects in early larval stages [34,38–40]. Moreover, mutation of the kinase domain of *nekl-2* or *nekl-3* leads to uniform molting defects indicating that the catalytic activity of the NEKLs is essential for the promotion of molting [34,39].

Molting is an essential feature for *C. elegans* to develop through its four larval stages and to reach adulthood. During molting, the old cuticle, an apical extracellular matrix (aECM) made up of proteins and lipids, detaches from the underlying epidermis and is broken down as a new cuticle is synthesized underneath. This process of aECM remodeling requires the trafficking of macromolecules required for the synthesis, degradation, and recycling of cuticle components [41,42]. We previously reported that NEKL-2 and NEKL-3 regulate clathrin-mediated endocytosis (CME) at the apical membrane of hyp7. Our findings indicated that loss of NEKL-2 or NEKL-3 leads to a defect in the uncoating of apical clathrin-coated vesicles, thereby preventing the flow of apical cargo through the endocytic pathway. Consistent with this, an apically expressed low-density lipoprotein–like receptor, LRP-1/Megalin, was observed to be trapped at or near the apical membrane in NEKL-depleted animals [34,43]. LRP-1 internalization by hyp7 is essential for molting, as it is required for the uptake of cholesterol, a substrate required for hormonal signaling pathways that coordinate the molting cycle [41,44,45].

The involvement of NEKLs in CME is also strongly supported by our genetic screens, which identified subunits of the clathrin adapter protein complex, AP2, as strong suppressors of both molting and trafficking defects in *nekl-2* and *nekl-3* mutants. Likewise, loss of FCHO-1, an allosteric activator of AP2, alleviates molting and trafficking defects in *nekl* mutants [43]. Our collective data indicate that NEKL-2 and NEKL-3 directly or indirectly promote clathrin uncoating and suggest that AP2 activity can also affect the uncoating process. However, a possible role for NEKLs in other aspects of intracellular trafficking was not explored.

In this study, we further investigated the role of NEKLs at different membranes and in different compartments within the endocytic network. Importantly, we found that NEKL-2 and NEKL-3 have distinct functions within different endosome types and are required for basolateral trafficking and cargo recycling. Importantly, we extended our studies to human cell lines and demonstrated that the human NEKL-3 orthologs, NEK6 and NEK7, also appear to function at multiple points within the endocytic pathway in mammals. These studies provide new insights into the biological roles of NEK/NEKL family members, which may have relevance to their roles in human development and disease.

## Results

### NEKL-2 and NEKL-3 colocalize with endosomal markers

Previously, we reported that NEKL-2 and NEKL-3 are specifically expressed and required within the large epidermal syncytium of *C. elegans*, hyp7 [34,38]. Moreover, both NEKL-2 and NEKL-3 are cytoplasmic and are localized to variably sized actin-rich puncta or aggregates, which are reminiscent of endosomal compartments [34,38]. To determine the subcellular compartments in which NEKLs may function, we carried out colocalization experiments using endogenously tagged NEKL strains, together with reporters for early ($P_{rab-5}$::GFP::RAB-5) and late ($P_{hyp7}$::GFP::RAB-7) endosomes [46–50]. In the case of NEKL-3, we observed robust colocalization of NEKL-3::mKate with GFP::RAB-5 and GFP::RAB-7 (~50% Manders' overlap with each marker), such that a majority of detectable NEKL-3 may be associated with early-to-late endosomes (Fig 1A–1F, 1A'–1F' and 1M). In contrast, whereas ~40% of NEKL-2::mKate colocalized with RAB-5::GFP, only ~10% colocalized with RAB-7::GFP, suggesting that NEKL-2 may play a greater role in early steps of endocytic trafficking (Fig 1G–1L, 1G'–1L' and 1M). We also

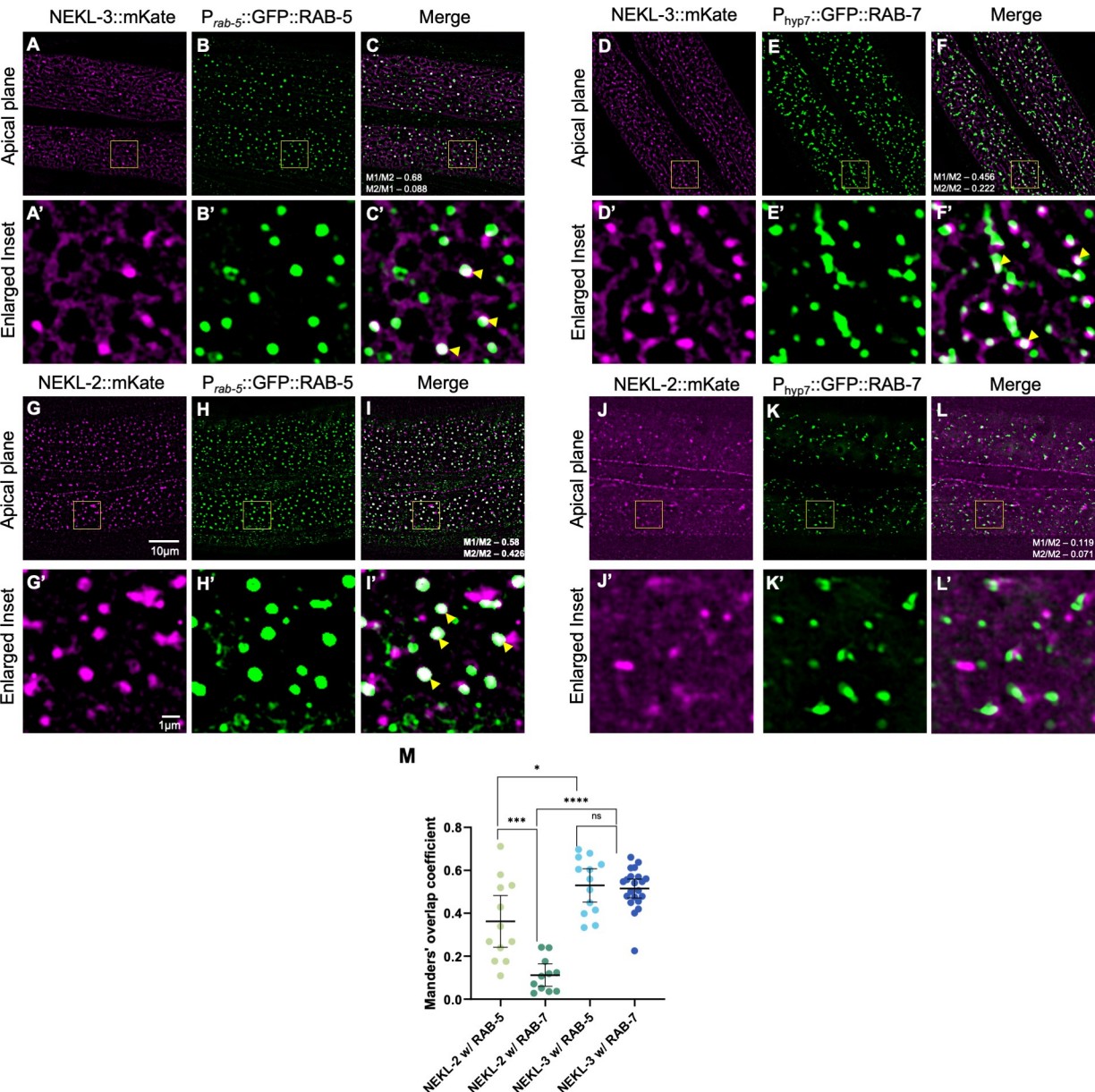

**Fig 1. Colocalization of NEKL-2 and NEKL-3 with endosomal markers in *C. elegans*.** Colocalization assays were carried out in adult worms expressing either NEKL-2::mKate or NEKL-3::mKate with endosomal markers $P_{rab-5}$::GFP::RAB-5 or $P_{hyp7}$::GFP::RAB-7. (A–C, A'–C', G–I, and G'–I') Representative confocal images of adult worms expressing either NEKL-3::mKate (A–C and A'–C') or NEKL-2::mKate (G–I and G'–I'), along with the early endosomal marker $P_{rab-5}$::GFP::RAB-5. (D–F, D'–F', J–L, and J'–L') Representative confocal images of adult worms expressing either NEKL-3::mKate (D–F and D'–F') or NEKL-2::mKate (J–L and J'–L') and late endosomal marker $P_{hyp7}$::GFP::RAB-7. Yellow arrowheads indicate overlap. Scale bar in A = 10 μm for A–L. Scale bar in A' = 1 μm for A'–L'. (M) Manders' coefficient was calculated for all the worms with the indicated backgrounds and plotted to determine the fraction of overlap between NEKL-2– or NEKL-3–positive pixels and the indicated markers. Error bars represent the 95% confidence intervals. p-Values were obtained by comparing means using an unpaired t-test: ****$p < 0.0001$, ***$p < 0.001$, *$p < 0.05$; ns, not significant ($p > 0.05$). Raw data are available in S1 File.

note that we previously reported the association of NEKL-2–MLT-2–MLT-4 with actin-rich puncta that form along the boundary of hyp7 and the seam cells, which do not appear to be endosomal and may represent adhesion junctions [38,39]. In addition, we have observed some diffuse reticulated signal from fluorescently tagged NEKL-3 and NEKL-2 in the cytoplasm of hyp7 (Fig 1), which could correspond in part to the endoplasmic reticulum [39].

The differences observed between NEKL-2 and NEKL-3 localization are consistent with our previous report that NEKL-2 and NEKL-3, along with their MLT co-partners, exhibit distinct localization patterns in hyp7 [39]. Consistent with this, we find that the Manders' overlap between fluorescently tagged NEKL-2 and NEKL-3 is <20% (S1 Fig), suggesting that the cellular functions of these kinases are distinct, despite having similar terminal molting-defective phenotypes. Notably, although a loss of NEKL-2 and NEKL-3 lead to defects in clathrin uncoating [43], neither showed strong colocalization with a marker for the AP2 clathrin-adapter protein complex, APA-2::mScarlet (S2 Fig). This suggests that the previously described impact of NEKLs on clathrin uncoating may be indirect or that levels of NEKLs at clathrin-coated vesicles are below detection. Notably, both NEKLs colocalize with RAB-5, which plays a role in the uncoating of clathrin-coated vesicles [51].

## NEKLs are required for normal endosome morphologies

Given the observed colocalization of NEKLs with endosomes, we next wanted to determine if the loss of NEKLs led to an observable defect in the morphologies of endosomal compartments. During the process of endocytosis, endosomes undergo maturation, which is accompanied by changes in shape, molecular composition, and function. In the case of early/sorting and recycling endosomes, tubular extensions extend and pinch off from larger vesicles, allowing proteins to be recycled to specific destinations such as the plasma membrane and trans-Golgi [52–55]. Disruptions to endosomal functions can lead to a wide variety of defects including alterations in cargo localization and abundance.

To avoid indirect consequences caused by molting defects, we assessed the roles of NEKLs on endosomal morphology using the auxin-inducible degron (AID) system, in which endogenously tagged NEKL-2::AID and NEKL-3::AID were depleted in day-1 adults after the addition of auxin [43,56,57]. Depletion of NEKL-2::AID led to an increase in the overall size of the early endosome compartment (based on total intensity and area measurements; $P_{rab-5}$::GFP::RAB-5) and, most strikingly, to an increase in the extent of tubulation (Fig 2A, 2B and 2D–2F). This phenotype suggests a requirement for NEKL-2 in the process of vesicle fission at early endosomes. In contrast, loss of NEKL-2 had more modest effects on the late endosome compartment ($P_{hyp7}$::GFP::RAB-7), consistent with the observed strong colocalization of NEKL-2 to early, but not late, endosomes (Fig 2G, 2H and 2J–2L). Specifically, although the mean number of vesicles was decreased in NEKL-2–depleted worms as compared with wild type, the morphology of late endosomes was not strongly affected, and the observed changes could be due in part to gross alterations in the early endosome caused by NEKL-2 depletion.

In the case of NEKL-3::AID, depletion led to strong defects at both early and late endosomes, consistent with the localization of NEKL-3 to both compartments. In the case of early endosomes, however, the effects of NEKL-3 loss differed markedly from those after loss of NEKL-2 (Fig 2B–2F). Rather than increased tubulation, loss of NEKL-3 led to increased variability in the size and shape of early endosomes accompanied by an increase in total signal intensity. Thus, while NEKL-2 and NEKL-3 both act at early endosomes, their roles in controlling early endosome morphology appear to be distinct. Depletion of NEKL-3 also led to a strong overall decrease in the abundance of late endosomes, which was associated with a decrease in total intensity, vesicle size, and changes in vesicle morphology (Fig 2G and 2I–2L). These defects may be due to a combination of the roles for NEKL-3 in directly regulating late endosomes and the potential secondary effects associated with its effects on early endosomes. We also note that even in the absence of auxin, NEKL-3::AID worms are smaller than wild-type worms and show some arrest due to molting defects, which may be attributable to a partial LOF caused by the AID tag [43].

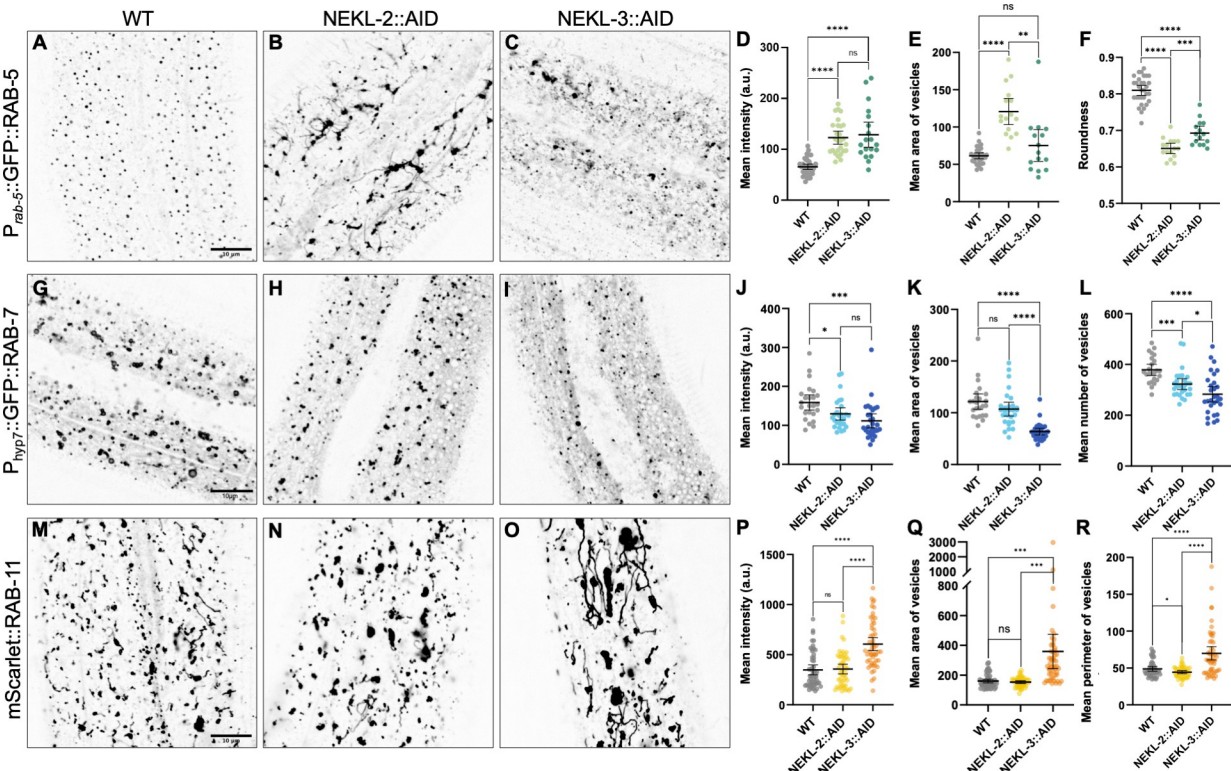

**Fig 2. Effects of NEKL-2 or NEKL-3 depletion on endosomal compartments in *C. elegans*.** (A–C, G–I, M–O) Confocal imaging was used to examine the effects of NEKL-2 (B, H, N) and NEKL-3 (C, I, O) loss relative to wild type (WT; A, G, M) in hyp7 of day-2 adult worms after auxin treatment. Representative images are shown. (A–C) Imaged worms expressed P$_{rab-5}$::GFP::RAB-5 in the indicated backgrounds. (D–F) The mean intensity (D), vesicle area (E), and the roundness of the puncta (F) were plotted for worms expressing P$_{rab-5}$::GFP::RAB-5. (G–I) Imaged worms expressed P$_{hyp7}$::GFP::RAB-7 in the indicated backgrounds. (J–L) The mean intensity (J), vesicle area (K), and the number of vesicles (L) were plotted for worms expressing P$_{hyp7}$::GFP::RAB-7. (M–O) Imaged worms expressed mScarlet::RAB-11, a marker for recycling endosomes, in the indicated backgrounds. (P–R) The mean intensity (P), vesicle area (Q), and perimeter of the vesicles (R) were plotted for individual worms expressing mScarlet::RAB-11. Area is in square pixels; perimeter is the number of pixels in the boundary of the object. Error bars represent the 95% confidence intervals. p-Values were obtained by comparing means using an unpaired t-test: ****p < 0.0001, ***p < 0.001, *p < 0.05; ns, not significant (p > 0.05). Raw data are available in S1 File.

We also examined the effects of NEKL loss on recycling endosomes, which were marked with endogenously (CRISPR) tagged RAB-11. Whereas NEKL-2::AID depletion had little or no effect on mScarlet::RAB-11 localization, we observed a strong increase in tubulation after loss of NEKL-3:AID, which was accompanied by an increase in total intensity and vesicle size (Fig 2M–2R). For technical reasons, we were unable to carry out colocalization studies between the NEKLs and either mScarlet::RAB-11 or GFP::RAB-11. Our observations, however, suggest that NEKL-3 may also function at recycling endosomes, possibly to control vesicle fission. In contrast to endosomal compartments, depletion of NEKL-2::AID and NEKL-3::AID had little or no effect on the morphology of the Golgi compartment, based on a P$_{hyp7}$::AMAN-2:: mNeonGreen marker (S3 Fig). Nevertheless, the total number of AMAN-2–positive vesicles was variably reduced in these backgrounds, with strongest effects shown for NEKL-3::AID (S3A–S3C Fig), which could be caused by reduced endosome-to-Golgi recycling. Consistent with an indirect effect, a multi-copy NEKL-3::mCherry reporter [34] showed very low levels of colocalization with AMAN-2:: mNeonGreen (S3D Fig). Collectively, our studies demonstrate a role for NEKL-2 and NEKL-3 in regulating endosomal morphologies and further suggest direct functions within endosomal compartments.

## NEKLs are required for basolateral cargo uptake by hyp7

We previously showed that NEKLs are required for CME at the apical membrane of hyp7 (Fig 1A and 1B) [44,45,58]. Specifically, depletion of NEKL-2 or NEKL-3 led to the accumulation of both clathrin and LRP-1, a low-density lipoprotein–like receptor, at the apical hyp7 surface [34,43,59,60]. Consistent with this, *C. elegans* LRP-1 endocytosis is dependent on clathrin and the clathrin adapter protein complex, AP2, analogous to findings in mammals [34,43,60–64]. To determine whether NEKL kinases are also required for trafficking at basolateral membranes of hyp7, we examined the internalization of two basolaterally localized BMP-family receptors. SMA-6 (Type I BMP receptor) and DAF-4 (Type II BMP receptor) form a heteromeric complex that binds to DBL-1 (BMP ligand), leading to the regulation of genes controlling *C. elegans* body size and male mating structures [65]. The internalization of SMA-6 depends on CME, as loss of AP2 function leads to the accumulation of SMA-6 at the basolateral membrane of intestinal cells [66]. In contrast, DAF-4 internalization does not require AP2, suggesting that it is endocytosed through a clathrin-independent mechanism [66].

To investigate possible roles for NEKLs in basolateral endocytosis, we expressed SMA-6::GFP and DAF-4::GFP using a hyp7-specific promoter and examined expression at the membrane of hyp7 adjacent to the seam cell (referred to hereafter as the lateral membrane) and at the membrane that forms the interior boundary of hyp7 (referred to hereafter as the basal membrane; Fig 3A and 3B). Notably, the mean intensity of $P_{hyp7}$::SMA-6::GFP was increased by ~1.3-fold or ~2.3-fold when NEKL-2 or NEKL-3 were depleted, respectively (Fig 3C–3E and 3H). Moreover, a strong increase (~4.8-fold) in the total level of $P_{hyp7}$::DAF-4::GFP was observed after NEKL-3::AID depletion (Fig 3F, 3G and 3I), suggesting that NEKL-3 may also be required for non-clathrin–dependent endocytosis or at the downstream junction of these pathways. In contrast, $P_{hyp7}$::DAF-4::GFP was not detectably affected by depletion of NEKL-2 (Fig 3H and 3J), suggesting that NEKL-2 may not be required for non-CME or may have a more limited role in endocytosis at the basolateral membrane relative to NEKL-3. Accumulation of these markers after NEKL-3 depletion was most evident at or near the basal hyp7 membrane, indicating a defect in plasma membrane uptake and/or processing through the early steps of endocytosis. In addition, some accumulation at or near the basolateral membrane with seam cells was also observed (S4 Fig). Together these results show that NEKL-2 and NEKL-3 play variable roles in the endocytosis of basolateral BMP receptors, although much stronger effects were observed for NEKL-3 in this process.

## NEKLs are required for cargo sorting

Membrane cargoes that are internalized by endocytosis are first delivered to sorting/early endosomes after which they may be recycled back to their original compartment(s) or routed to the lysosome for degradation. Previous studies have characterized the recycling routes for two conserved cargo proteins, MIG-14/Wntless and TGN-38/TGN38, both of which originate from the trans-Golgi [67–71]. MIG-14 is a transmembrane protein that binds to Wnt ligands in the Golgi and is responsible for delivering Wnts to the plasma membrane for secretion. Once at the plasma membrane, MIG-14 is endocytosed and recycled back to the Golgi through a retrograde pathway that requires the retromer complex [72]. In the absence of retromer function, MIG-14 is missorted to the lysosome and is degraded [67,68]. Likewise, TGN-38 is a trans-Golgi resident that is also recycled from the plasma membrane to the trans-Golgi by a retromer-associated retrograde pathway [67,73,74].

In wild-type adults, both $P_{hyp7}$::MIG-14::GFP and $P_{hyp7}$::TGN-38::GFP markers were observed in punctate structures of varying size throughout the cytoplasm, with some accumulation occurring at the lateral surface (Fig 4A and 4E). After depletion of NEKL-3::AID, $P_{hyp7}$::

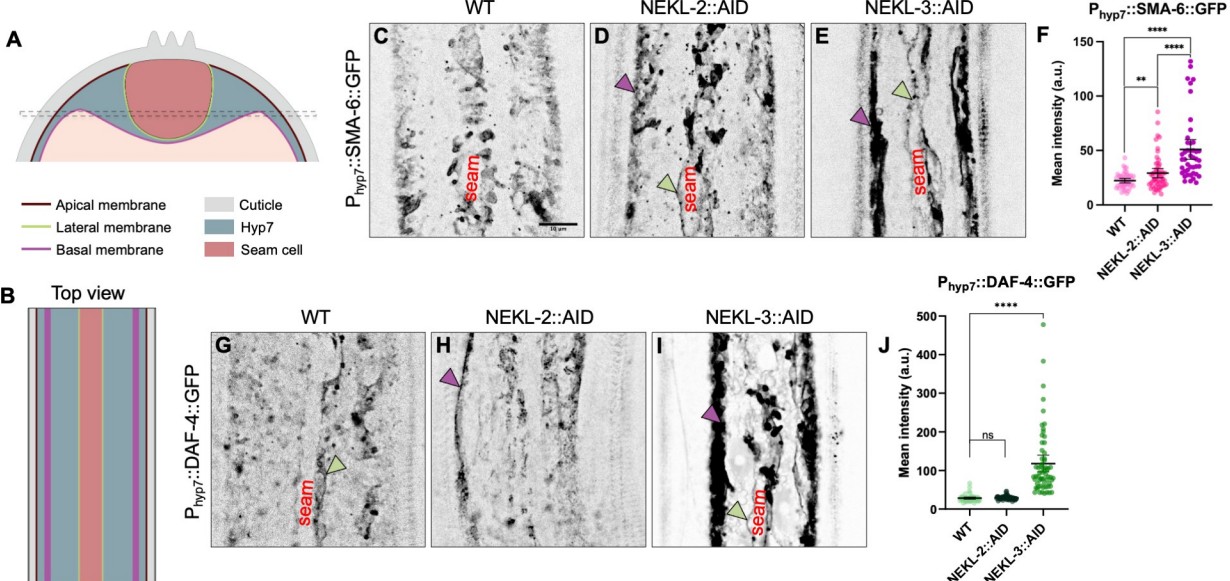

**Fig 3. Effects of NEKL-2 or NEKL-3 depletion on basolateral cargoes in *C. elegans*.** (A) Cross-sectional view of an adult worm depicting the position of the hyp7 syncytium (blue grey) and seam cell syncytium (red). Specific membrane domains of hyp7 are indicated: apical, facing externally directly underneath the cuticle (brown); basal, facing internally (purple); lateral, facing the sides and bottom of the seam cell (green). Dashed lines indicate the medial imaging plane used to acquire the images. (B) Top-down view of the long axis of the worm's body indicating the positions of the hyp7 and seam cell syncytium along with indicated membranes as in A. (C–E) Representative confocal images of $P_{hyp-7}$::SMA-6::GFP expression in auxin-treated wild-type (C), *nekl-2::aid* (D), and *nekl-3::aid* (E) day-2 adults. (F, G) Representative confocal images of $P_{hyp-7}$::DAF-4::GFP expression in auxin-treated wild-type (G), *nekl-2::aid* (H), and *nekl-3::aid* (I) day-2 adults. Green and purple arrowheads (D,E,G) indicate seam and basal membranes (where detectable), respectively. Note that it is often not possible to identify the precise lateral/basolateral boundaries of hyp7 in the SMA-6::GFP and DAF-4::GFP lines in wild type because of the curved nature of the lateral membrane and because of low levels of these cargos marking the membrane. Scale bar in C = 10 μm for C–E, F, and G. (F, J) Mean intensity values for $P_{hyp-7}$::SMA-6::GFP (F) and $P_{hyp-7}$::DAF-4::GFP (J) expression were plotted for individual adults. The two highest datapoints in *nekl-3::aid*; DAF-4::GFP (3J; 615 and 1233) were omitted for clarity of presentation. Error bars represent the 95% confidence intervals. Statistical significance was determined using a two-tailed, unpaired t-test: **p < 0.01, ****p < 0.0001. Raw data are available in S1 File.

MIG-14::GFP expression was decreased ~2.7-fold, whereas depletion of NEKL-2::AID resulted in at most a subtle decrease in total $P_{hyp7}$::MIG-14::GFP levels (~1.2-fold; p = 0.09) (Fig 4A–4D). In the case of $P_{hyp7}$::TGN-38::GFP, marker expression was decreased by ~2.8-fold after depletion of either NEKL-2::AID or NEKL-3::AID (Fig 4E–4H). Such findings further highlight differences between NEKL-2 and NEKL-3 with respect to their effects on specific cargoes and suggest that MIG-14::GFP and TGN-38::GFP may be missorted to the lysosomal degradative pathway after loss of NEKLs.

To test if NEKL depletion leads to the aberrant degradation of cargoes, we used CRISPR methods to introduce a partial loss-of-function mutation into *cup-5/trpml1*, which encodes a conserved channel protein required for the maturation of lysosomes and whose loss leads to reduced lysosomal function [75–77]. Notably, reduction of CUP-5 activity led to an ~1.5-fold increase in the total expression of $P_{hyp7}$::TGN-38::GFP in both NEKL-2::AID and NEKL-3::AID depleted strains (Fig 4I–4K). Moreover, accumulation of $P_{hyp7}$::TGN-38::GFP was detected in moderate-to-large internal structures in these strains, which may correspond to defective lysosomes. Together these results indicate that, in addition to functioning in membrane cargo uptake, NEKL-2 and NEKL-3 are also required for the correct sorting and recycling of internalized cargo. We note that loss of *cup-5* in an otherwise wild-type background also led to the accumulation of $P_{hyp7}$::TGN-38::GFP, suggesting that some portion of TGN-38 is normally delivered to the lysosome in wild-type animals and that it's degradation depends on lysosomal function (S5 Fig). TGN-38 accumulations, however, appeared larger in *cup-5*

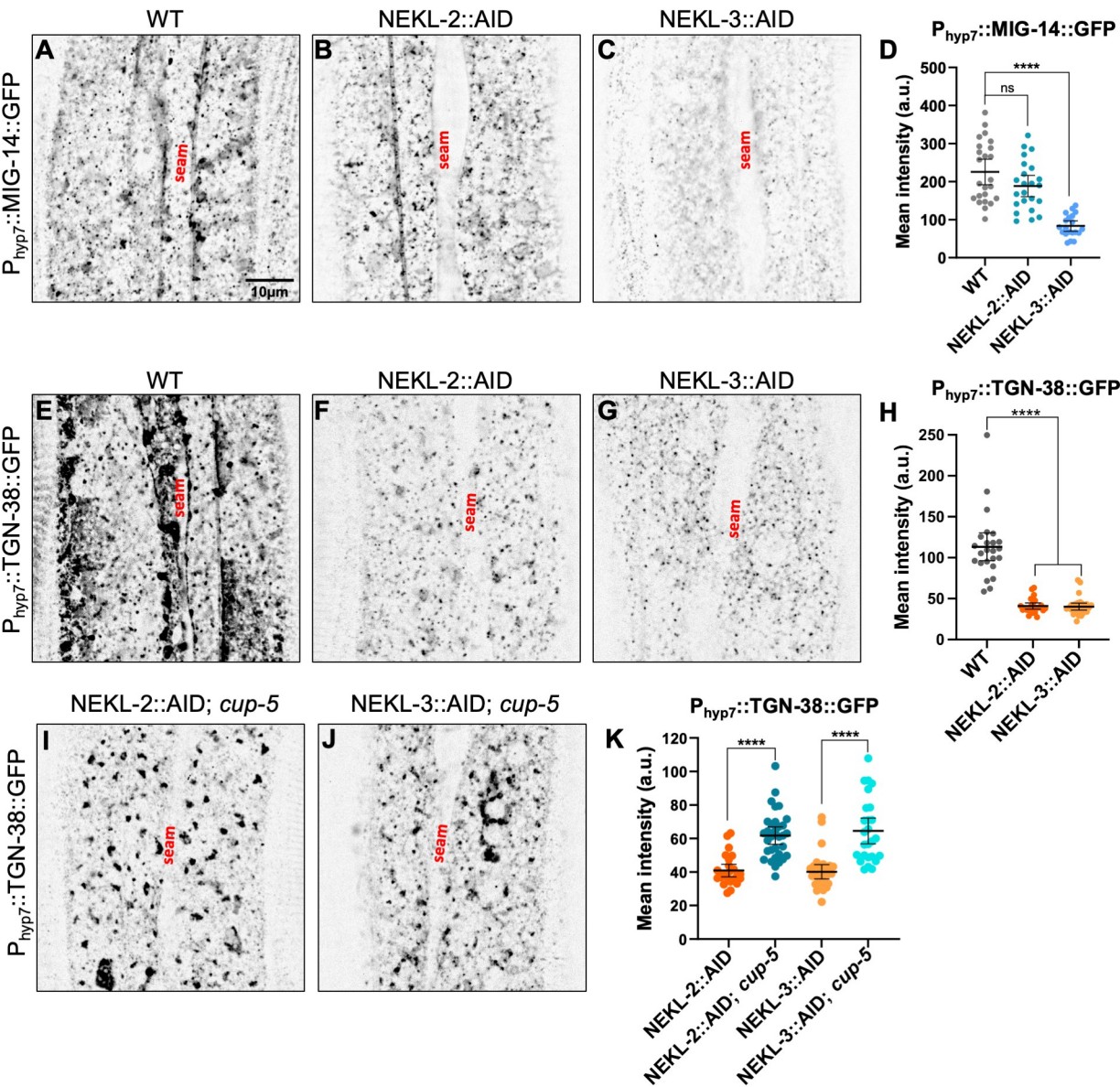

**Fig 4. Effects of NEKL-2 or NEKL-3 depletion on cargo sorting in *C. elegans*.** (A–C, E–G, and I–J) Representative confocal images of P$_{hyp-7}$::MIG-14::GFP (A–C) and P$_{hyp-7}$::TGN-38::GFP (E–G and I–J) expression within hyp7 in auxin-treated wild-type (A, E), NEKL-2::AID (B, F), NEKL-3::AID (C, G), NEKL-2::AID; *cup-5* (I), and NEKL-3::AID; *cup-5* (J) day-2 adults. Scale bar in A = 10 µm for A–C, E–G, I, and J. (D, H) Mean pixel intensity values of P$_{hyp-7}$::MIG-14::GFP and P$_{hyp-7}$::TGN-38::GFP expression for individual worms. Error bars represent the 95% confidence intervals. Statistical significance was determined using a two-tailed, unpaired t-test: ****p < 0.0001; ns, not significant (p > 0.05). Raw data are available in S1 File.

single mutants relative to NEKL::AID; *cup-5* strains (Figs 4 and S5). One possible explanation for this difference is that NEKL depletion may directly or indirectly affect lysosomal morphology or lysosome abundance.

Depletion of NEKL-3::AID but not NEKL-2::AID also led to a reduction in the levels of P$_{hyp7}$::MIG-14::GFP at the hyp7 membrane surrounding the ALM and PLM neurons, which run longitudinally through hyp7 along the anterior-posterior axis (S6 Fig). Altogether our results indicate that loss of NEKLs leads to a range of defects affecting endosomal compartments as well as cargo internalization and sorting. Moreover, although *nekl-2* and *nekl-3*

mutants appear superficially identical with respect to molting defects, they act in largely distinct subcellular compartments and lead to different subcellular phenotypes when depleted.

## The mammalian homologs of NEKL-3, NEK6 and NEK7, are required for normal endocytic trafficking in human cells

Human NEK6 and NEK7 are both ~70% identical and ~85% similar to NEKL-3 and can rescue molting and trafficking-associated defects when expressed in *C. elegans nekl-3* mutants [43]. Nevertheless, a systematic examination of potential roles for NEK6 and NEK7 in intracellular trafficking has not been undertaken. Using siRNA approaches, we were able to substantially reduce protein levels of NEK6 and NEK7 within 48 h of oligonucleotide transfection into cell lines (Fig 5A). As a first measure, we examined the effects of NEK6 and NEK7 knockdown on the cation-independent mannose 6-phosphate receptor (M6PR) in HeLa cells grown at steady state. M6PR is responsible for the delivery of mannose 6-phosphate–tagged lysosomal hydrolases from the trans-Golgi network (TGN), and thus cycles rapidly between the TGN and late endocytic organelles. Accordingly, M6PR is highly sensitive to disruptions in endocytic trafficking, which lead to alterations in its distribution.

Relative to control cells, the mean fluorescence intensity of M6PR-containing endosomes showed a significant decrease (~3-fold) in both NEK6 and NEK7 knockdown cells (Fig 5B–5G and 5K). These findings suggest that, analogous to observations for MIG-14 and TGN-38 in *C. elegans*, M6PR may be aberrantly targeted for lysosomal degradation after loss of the NEKL-3 orthologs. In addition, the distribution of M6PR in NEK6/7-depleted cells was altered, with M6PR showing a 2- to 3-fold increase in the area of dispersal (Fig 5H–5J and 5L). Although this increased M6PR dispersal might reflect slightly larger cell sizes in the NEK-depleted cells, M6PR appears to be more homogeneously distributed in the knockdown cells, an effect that has been associated with defects in endosome-to-Golgi transport via the retromer complex [78]. These data imply that both NEK6 and NEK7 are involved in the regulation of endosome-to-Golgi retrograde trafficking.

Given that loss of NEKLs led to a major impact on endosomes, we next assessed the impact of NEK6 and NEK7 depletion on EEA1, a well-characterized early endosomal protein that binds to endosomal PI3P via its FYVE domain [79–81]. Upon siRNA-mediated depletion of NEK6 or NEK7 (Fig 6A) and immunostaining with antibodies against EEA1, endosomal size was increased in cells depleted of NEK6 or NEK7, with those lacking NEK7 displaying especially large ring-like structures decorated by EEA1 (Fig 6B–6G and 6H). In contrast, the number of EEA1 endosomes was not significantly altered upon depletion of either NEK6 or NEK7 (Fig 6I). As a further test, we performed NEK6/7 siRNA knockdown in glioblastoma cells and observed an increase in the size of EEA1-marked vesicles after depletion of NEK7 but not NEK6 (S7 Fig), suggesting cell type–specific requirements for NEK functions. Together, these data suggest a role for NEK6 and NEK7 at endosomes, potentially in the regulation of fusion and/or fission events.

To further examine the role of NEK6 and NEK7 at endosomes, we analyzed the effect of their depletion on tubular recycling endosomes marked by the protein MICAL-L1 [82–85]. After depletion of NEK6 or NEK7 with siRNA oligonucleotides (Fig 7A), HeLa cells were fixed and immunostained with antibodies against endogenous MICAL-L1. Inhibition of NEK6 or NEK7 led to an increase in the MICAL-L1 tubular endosomal surface area relative to controls (Fig 7B–7E). Similar observations were observed for glioblastoma cells, with depletion of either NEK6 or NEK7 leading to increased tubulation (S8 Fig). Together, these data indicate roles for NEK6 and NEK7 in endosomal morphology and function in mammalian cells and support the possibility of their involvement in fusion and/or fission activities.

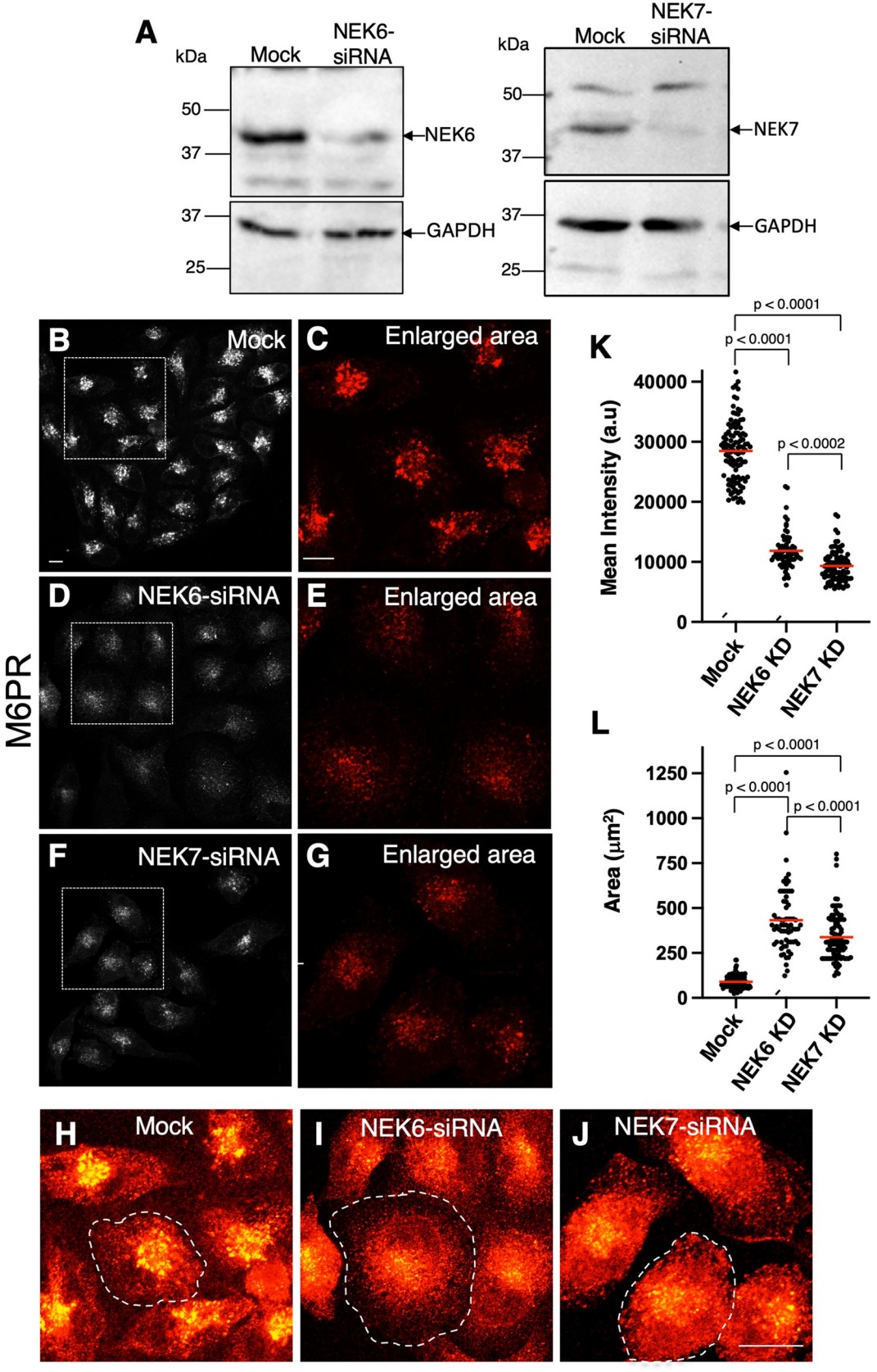

**Fig 5. Effects of NEK6 or NEK7 depletion on mannose 6-phosphate receptor trafficking in human cells.** (A) siRNA knockdown of NEK6 and NEK7 was validated by western blotting of HeLa cells that were mock-transfected or transfected with oligonucleotides specific for NEK6 (left panel) or NEK7 (right panel). GAPDH was used as the loading control (lower panels). (B–G) Mock-transfected cells (B and enlarged area in C), NEK6 siRNA–transfected cells (D and enlarged area in E), or NEK7 siRNA–transfected cells (F and enlarged area in G) were plated on coverslips and immunostained with antibodies against mannose 6-phosphate receptor (M6PR). (K, L) The mean intensity of M6PR immunostaining (K) and the mean area of M6PR distribution (L) of individual cells in mock-transfected, NEK6 siRNA–transfected, and NEK7 siRNA–transfected cells. (H–J) Saturated and zoomed micrographs demonstrating the distribution and intensity of mannose 6-phosphate receptor immunostaining in mock-transfected (H), NEK6 siRNA–transfected (I), and NEK7 siRNA–transfected cells (J). The dashed lines indicate individual cells. Scale bar in B = 10 μm for B, D, and F; scale bar in C = 10 μm for C, E, and G; scale bar in J = 10 μm for H–J. Statistical significance was determined using Student's unpaired t-test. Raw data are available in S1 File.

## Discussion

### NIMA kinases are required for proper endosome morphology

In this study we have demonstrated that *C. elegans* NEKLs and human NEKs are important for maintaining the proper morphology of endosomal compartments. Loss of NEKL-2 led to enlarged early endosomal compartments with noticeably longer tubular extensions. These changes could indicate a failure of early endosomes to undergo fission, an expansion of early endosomes caused by abnormal cargo retention, or both. Notably, the effect of NEKL-3 loss on early endosomes was distinct from that of NEKL-2, as no increase in tubular extensions was observed. The shape of early endosomes, however, became irregular, and the mean intensity of the early endosomal marker GFP::RAB-5 was increased. This suggests that NEKL-2 and NEKL-3 have distinct roles in regulating early endosome morphology. Notably, siRNA-mediated knockdown of human NEK6 or NEK7 also led to an increase in the size of EEA-1–positive early endosomes in HeLa cells. However, whereas knockdown of NEK7 led to an increase in the size of early endosomes in glioblastoma cells, inhibition of NEK6 did not show this effect, suggesting that NEK6 and NEK7 may regulate early endosomal morphology in a cell type–specific manner.

Consistent with their distinct localization patterns, depletion of NEKL-3 had much stronger effects on late endosomal compartments than did loss of NEKL-2. We note, however, that effects on late endosomes could be due in part to perturbations of early endosomes caused by loss of NEKL-2 and NEKL-3. For technical reasons, we could not determine the expression of NEKLs in recycling endosomes. However, an increase in membrane tubulation was observed with recycling endosomes in hyp7 after loss of NEKL-3 but not NEKL-2. Collectively, these results indicate that NEKL-2 and NEKL-3 have important but distinct roles in regulating endosomal morphology.

### NIMA kinases may broadly affect cargo uptake and recycling

We examined two Golgi residents, MIG-14 and TGN-38, to better understand how NEKLs may influence the endocytic recycling process. We observed a dramatic reduction in the MIG-14 and TGN-38 signal after depletion of NEKL-3. In contrast, loss of NEKL-2 led to a strong reduction in TGN-38, but only minimal effects on MIG-14. These findings suggest that specific cargoes are variably missorted to the lysosomal degradative pathway when NEKL-2 or NEKL-3 is depleted, as was previously reported for mutants affecting actin assembly and the sorting nexin, SNX3 [67,68]. Consistent with this, expression of TGN-38 was rescued when lysosomal function was inhibited in NEKL::AID strains. Interestingly, defects consistent with abnormal recycling were also observed for M6PR when NEK6 and NEK7 were depleted in HeLa cells, consistent with M6PR being incorrectly targeted to lysosomes. Together these findings suggest that NIMA family kinases may have conserved functions in the sorting of cargoes from early endosomes.

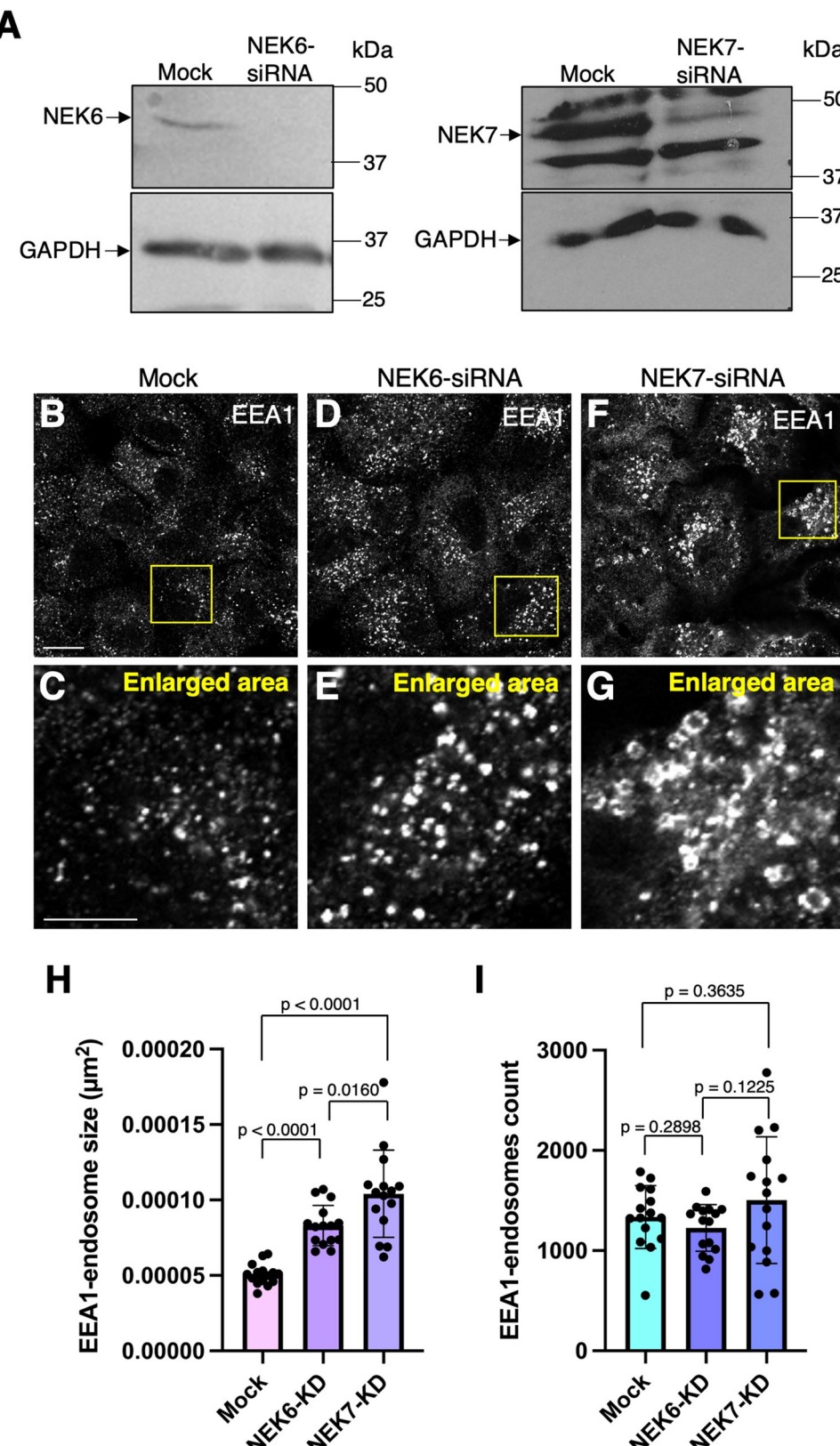

**Fig 6. Effects of NEK6 or NEK7 depletion on EEA1-positive sorting endosomes in human cells.** (A) siRNA knockdown of NEK6 (left panel) and NEK7 (right panel) in HeLa cells was confirmed by western blotting (B–G)

Mock-transfected cells (B and enlarged area in C), NEK6 siRNA–transfected cells (D and enlarged area in E), or NEK7 siRNA–transfected cells (F and enlarged area in G) were plated on coverslips and immunostained with antibodies against the early/sorting endosome marker protein EEA1. Yellow boxes in B, D, and F indicate the area of higher magnification shown in C, E, and G, respectively. (H) The mean size of EEA1-containing endosomes in mock-transfected, NEK6 siRNA–transfected, and NEK7 siRNA–transfected cells. (I) The number of EEA1-containing endosomes in mock-transfected, NEK6 siRNA–transfected, and NEK7 siRNA–transfected cells. Scale bar in B = 10 μm for B, D, and F; Scale bar in C = 5 μm for C, E, and G. Statistical significance was determined using Student's unpaired t-test. Raw data are available in S1 File.

Previously, we reported that NEKL-2 and NEKL-3 regulate endocytosis at the apical membrane of hyp7 [43]. In this study, we found that the internalization of a basolateral membrane cargo, SMA-6, was also affected in worms with NEKL-2 or NEKL-3 depletion. Our data also suggest that NEKL-3 may have a role in clathrin-independent endocytosis, as loss of NEKL-3 led to an accumulation of a putative clathrin-independent cargo, DAF-4, near the basolateral surface. One potential caveat to this interpretation is that in the presence of ligand, SMA-6 and DAF-4 would be predicted to form a complex [86], and thus SMA-6 or DAF-4 could potentially affect each other's uptake or retention. For example, membrane retention of endogenous SMA-6 could impact the localization and uptake of DAF-4::GFP. However, SMA-6 is expressed at very low levels [87,88], and thus it seems unlikely that endogenous plasma-membrane associated SMA-6 could indirectly lead to the strong membrane/juxta-membrane retention of the more highly expressed DAF-4::GFP. Altogether, these findings point to roles for NEKL-3 in both CME and non-CME.

We note that neither SMA-6 nor DAF-4 appeared to be incorrectly targeted to lysosomes after NEKL depletion, which differs from what we observed for MIG-14 and TGN-38. We speculate that SMA-6 and DAF-4 may be trapped at or near the basolateral surface in NEKL-3::AID depleted worms, as our previous study failed to detect clathrin accumulation at basolateral membranes of hyp7 [43]. Moreover, we failed to detect GFP::RAB-5 accumulation at or near basolateral surface in NEKL-3::AID depleted worms, suggesting that the basolateral accumulation of SMA-6 and DAF-4 is not within a RAB-5–marked early endosomal compartment. This buildup SMA-6 and DAF-4 could be due plasma membrane accumulation, internalized juxta-membrane vesicles, or a combination of the two.

## What are the core functions of NEKLs in intracellular trafficking?

In a previous study, we reported that NEKL-2 and NEKL-3 regulate CME at the apical membrane and that they promote the uncoating of clathrin from internalized vesicles [43]. Our current study, however, suggests that the primary sites for trafficking regulation by NEKL-2 and NEKL-3 may be endosomes, and we failed to observe strong colocalization of NEKL-2 or NEKL-3 to AP2–decorated clathrin coated pits and vesicles. Thus, the clathrin-uncoating defects we previously observed could in part be an indirect consequence of endosomal defects. In this context, it is worth noting that RAB-5, which colocalized with both NEKL-2 and NEKL-3, partially localizes to clathrin-coated pits and promotes clathrin and AP2 uncoating [51,89]. Moreover, human RAB5 has been detected at both early endosomes and clathrin-coated vesicles [46,90]. Thus, it remains possible that low levels of NEKL-2 and NEKL-3 may exert direct effects on clathrin uncoating, possibly in conjunction with RAB-5.

The effects we observed on multiple cargoes and compartments indicate that NEKL-2 and NEKL-3 may have widespread functions in intracellular trafficking. Likewise, effects on multiple compartments were observed after human NEK6 and NEK7 knockdowns. This could be due to NIMA kinases regulating distinct targets at different locations within the endocytic network. Alternatively, NIMAs may act on a smaller number of targets, which in turn have widespread functions in trafficking.

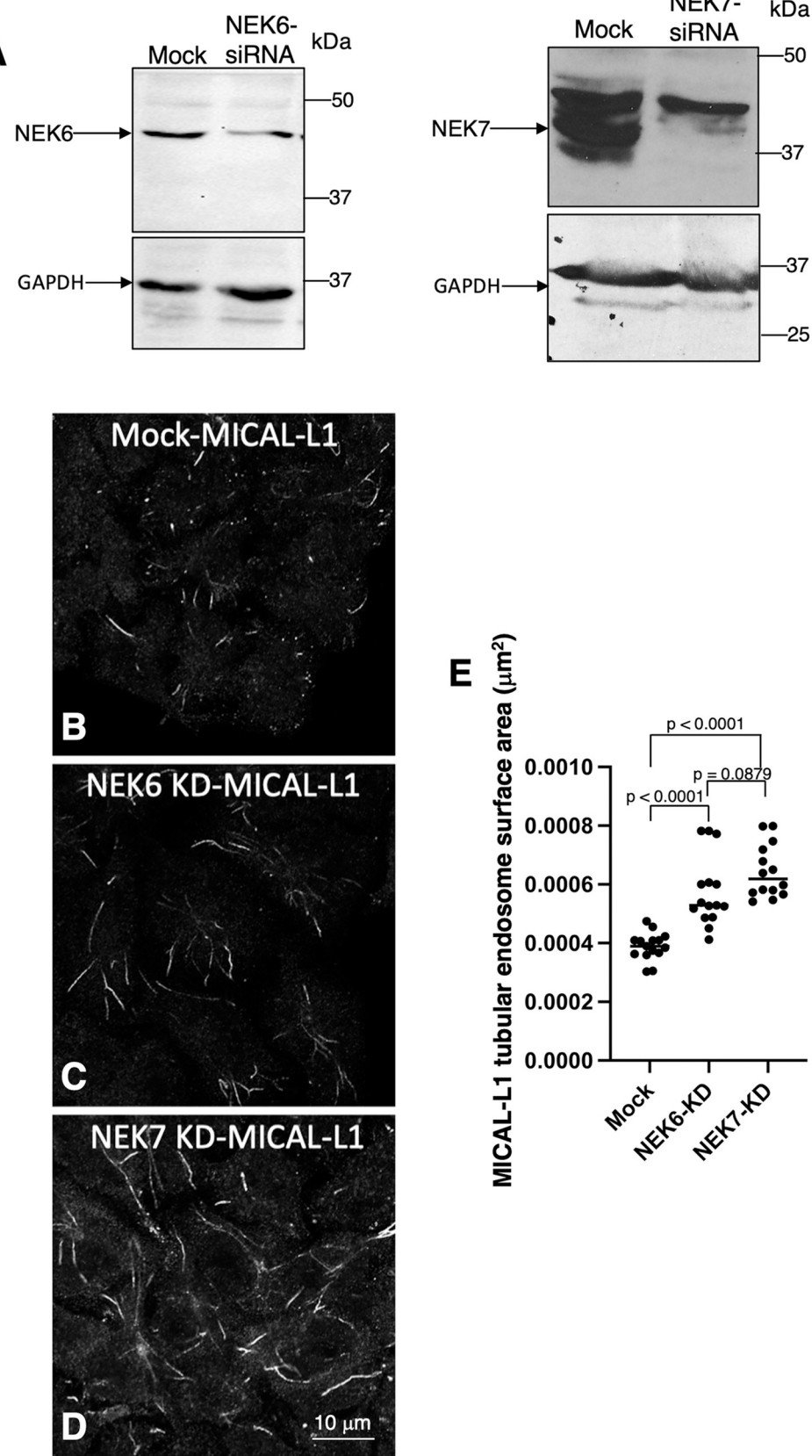

**Fig 7. Effects of NEK6 or NEK7 depletion on MICAL-L1–containing tubular recycling endosomes in human cells.**
(A) siRNA knockdown of NEK6 (left panel) and NEK7 (right panel) was validated by western blotting of HeLa cells.
(B–D) Mock-transfected cells (B), NEK6 siRNA–transfected cells (C), or NEK7 siRNA–transfected cells (D) were
plated on coverslips and immunostained with antibodies against the tubular recycling endosome marker protein
MICAL-L1. (E) The surface area of MICAL-L1–containing endosomes in mock-transfected, NEK6 siRNA–
transfected, and NEK7 siRNA–transfected cells. Scale bar in D = 10 μm for B–D. Statistical significance was
determined using Student's unpaired t-test. Raw data are available in S1 File.

In this latter category, one compelling model is that NIMAs could regulate endosomal
actin. Actin polymerization is deployed throughout endocytosis to provide mechanical forces
needed to bend membranes, facilitate vesicle fission, and promote short-range vesicle trans-
port [91–95]. For example, actin filaments are enriched at sorting endosomes and promote the
fission of tubular extensions [52,96]. Actin assembly also plays a role in the sorting of macro-
molecules from early endosomes and their transport to the trans-Golgi [97,98]. Consistent
with this idea, our previous studies showed that actin organization was strongly disrupted in
*nekl* mutants and that *nekl* molting defects can be suppressed by loss of function in the con-
served actin regulator CDC-42, along with one of its effectors, SID-3/ACK1/2 (activated
CDC42 kinase) [38]. Moreover, activated (GTP-bound) CDC-42 was strongly upregulated in
*nekl* mutants [38]. Interestingly, based on protein-interaction studies, CDC42 was reported to
interact with both NEK6 and NEK7 and could therefore connect the NEKL–MLT network to
actin polymerization and to their roles throughout the endocytic network. Future studies will
seek to characterize the functional link between NIMA kinases and the actin cytoskeleton,
along with their specific roles in regulating endocytosis.

## Human NEKs in endocytosis and cancer

Although human homologs of NEKL-3 (NEK6, NEK7) and NEKL-2 (NEK9) have largely been
studied for their roles in cell division, several studies have hinted at potential functions in endo-
cytosis [99–104]. A high-throughput siRNA-based screen of mammalian protein kinases indi-
cated that siRNA-mediated knockdown of NEK6 and NEK7 strongly disrupts CME in HeLa
cells [99]. Moreover, genome-wide siRNA studies showed that knockdown of multiple NEK
kinases, including NEK6, resulted in the abnormal uptake of endocytic cargoes [101]. A more
recent study identified NEK6 as a potential downstream target of NRP-1, which is required for
transferrin endocytosis by *Trypanosoma brucei* [105]. Moreover, proteomic studies of NEK6
and NEK7 identified interactions with several trafficking components including alpha and beta
subunits of the AP2 adaptor complex (NEK7) [102,103]. Such reports are consistent with our
current findings that NEK6 and NEK7 play roles in endocytic trafficking in human cells. Lastly,
our ability to rescue both molting and trafficking defects through the expression of human
NEKs in *C. elegans nekl* mutants suggests that these functions are conserved [43].

Finally, our studies provide evidence that NEKLs may affect two highly conserved signaling
pathways, BMP and Wnt, which are misregulated in many human cancers [106–108]. Previous
studies have described the role of endocytosis in cancers including how endocytosis adapts to
the needs of cancer cells, a phenomenon known as "adaptive CME" [109–111]. For example,
upregulation of clathrin light-chain (CLCb) leads to an increased rate of CME; this in turn
alters the trafficking of oncogenic epidermal growth factor receptor (EGFR) and promotes
cancer cell migration and metastasis [112]. Thus, it is possible that NIMA kinases, which are
overexpressed in many tumors, could be driving cancer formation in part through their roles
in trafficking as well as by their previously described functions in mitosis [19,113–117]. Future
studies will elucidate how NIMA kinases control trafficking, which will aid in understanding
their involvement in human disease.

## Materials and methods

### Strains and maintenance

*C. elegans* strains were maintained according to standard protocols [118] and were propagated at 22°C unless stated otherwise. Strains used in this study are listed in S1 Table. Loss-of-function alleles of *cup-5* (*fd395* and *fd397*) were generated by the CRISPR-Cas9 protocol described by Ghanta et al. [119]. Sequences for the sgRNA and repair template are given below. Capital letters represent the altered or inserted nucleotides.

sgRNA: 5′-aacgatgcgcttattatcat-3′

Repair Template: 5′-aacttaaatttttataaaaattacctgaaactcgatggatattttttgcaacGGTCTCaatgataa-taagcgcatcgttgaccacaatcatcacataccacaaattcag-3′

### Reporter strain construction

Plasmids for *C. elegans* hyp7-specific expression used the promoter from *semo-1/Y37A1B.5* (P$_{hyp-7}$) [120]. Vector details are available upon request. Cloning was performed using the Gateway system (Invitrogen) and modified versions of hygromycin-resistant and miniMos-enabled vector pCFJ1662 (gift from Erik Jorgensen, University of Utah; Addgene #51482). pDONR221 entry vectors containing coding regions for *rab-5*, *rab-7*, *mig-14*, *tgn-38*, *aman-2*, *daf-4*, and *sma-6* were transferred into hyp7 destination vectors with the Gateway LR clonase II reaction to generate C-/N-terminal fusions. Single-copy integrations were obtained using miniMos technology [121].

### Image acquisition

Confocal fluorescence images in Figs 1–4 and S1 and S3 and S5 were acquired using an Olympus IX83 inverted microscope with a Yokogawa spinning-disc confocal head (CSU-W1). z-Stack images were acquired using a 100×, 1.35 N.A. silicone oil objective. cellSense 3.1 software (Olympus Corporation) was used for image acquisition. Fluorescence images in S2 Fig were acquired using an Olympus IX81 inverted microscope with a Yokogawa spinning-disc confocal head (CSU-X1). MetaMorph 7.7 software was used for image acquisition. z-Stack images were acquired using a 100×, 1.40 N.A. oil objective.

Confocal images for Figs 5–7 and S6 and S7 were acquired using an Olympus IX83 inverted microscope with a Yokogawa spinning-disc confocal head (CSU-W1) or with a Zeiss LSM 800 microscope with a 63×, 1.40 N.A. objective with appropriate filters.

### Image analysis

Mean intensity (measured in arbitrary units, a.u.), quantification of various shape parameters, and the colocalization analysis were performed using Fiji software [122]. To quantify mean intensity (Figs 2D, 2J, 2P, 3H, 3I, 4D, 4H, 4K and S3B) for a z-plane of interest, rolling ball background subtraction was performed (radius = 50 pixels), and the polygon selection tool was used to choose the region of hyp7 in which the mean intensity was quantified.

To quantify shape parameters, a z-plane of interest was selected, and the minimum filter (radius = 10.0 pixels) was applied to the raw image. The filter-applied image was then subtracted from the raw image using the image calculator function. Images were then thresholded, and the Despeckle function was used to remove noise corresponding to 1 pixel in size. Finally, various shape parameters were quantified using the Shape Descriptor plugin.

For colocalization studies, the raw z-stack images were deconvoluted using the Wiener deconvolution algorithm (cellSense 3.1 software). The desired z-plane was extracted from both deconvoluted and raw z-stack images. Then, the Gaussian filter (radius = 10 pixels) was

applied to the deconvoluted image, which was subtracted from the original deconvoluted image using the image calculator function. Next, these subtracted images were thresholded to obtain binary images to be used as masks. These binary masks were combined using the "AND" Boolean operation to the background-subtracted (rolling ball algorithm; radius = 50 pixels) raw images. Using the polygon tool, the region of hyp7 was selected in the combined images, and coloc2 plugin was used to calculate Manders' coefficient.

The mean area and intensity of fluorescence in HeLa cells Fig 5K and 5L was obtained with Zeiss LSM Zen software, after outlining a region of interest. EEA1-positive endosome size (Fig 6H) was measured using Imaris software, and MICAL-L1 tubular endosome surface area (Fig 7E) was quantified using ImageJ software., or the number of MICAL-L1 tubular recycling endosomes were counted manually (S8 Fig), since number and area are generally correlated. Endosome size in the human cell studies was quantified using NIH ImageJ. Size parameters were set for 0 –infinity. Brightness parameters were selected to eliminate recognition of background by ImageJ's particle counter while optimizing selection of true positive fluorescent pixels. Statistical analyses was performed in Prism by first testing the assumption of normal distribution with the D'Agostino and Pearson normality test. Statistical significance was then calculated with an unpaired two-tailed t-test.

## Auxin treatment

Auxin (indole-3-acetic acid) was purchased from Alfa Aesar. A 100× stock auxin solution (0.4 M) was made by dissolving 0.7 g of auxin in 10 ml of 100% ethanol. A mixture of 25 µl of stock auxin solution and 225 µl of autoclaved deionized water was added to each plate containing day-1 adult worms.

## Antibodies

Rabbit antibodies against NEK6 were obtained from Mybiosource (Cat# MBS94186), rabbit antibodies against NEK7 were obtained from Novus (Cat# 31110), mouse horseradish peroxidase (HRP)-conjugated antibodies against GAPDH were obtained from Proteintech (Cat# HRP-60004), rabbit antibodies against EEA1 were obtained from Cell Signaling (Cat# 3288), and mouse polyclonal antibodies against MICAL-L1 were obtained from Novus (Cat# H00085777-B01P).

## Immunoblotting

HeLa cells were lysed in lysis buffer containing 50 mM Tris-HCl (pH 7.4), 150 mM NaCl, 1% NP-40, and 0.5% deoxycholate. Proteins from the lysates were separated by SDS-PAGE, transferred to nitrocellulose filters, and immunoblotted with antibodies using standard methods.

## siRNA knockdown

Cells were transfected with siRNA oligonucleotides from Sigma (600 nM NEK6 600 or 100 nM NEK7) using Dharmafect transfection reagent (Dharmacon) for 48 h, prior to validation of knockdown by immunoblotting using GAPDH as the loading control as described above.

## Statistics

All statistical tests were performed using software from Prism GraphPad.

## Supporting information

**S1 Fig. Colocalization between NEKL-2 and NEKL-3.** Colocalization assays were carried out in adult worms expressing both NEKL-2::mNeonGreen and NEKL-3::mKate. (A–C, A'–C', D–

F, and D'–F') Representative images of the apical region of hyp7 in adult worms expressing NEKL-2::mNeonGreen and NEKL-3::mKate (A–C and A'–C') as well as the medial plane of hyp7 (D–F and D'–F'). Here the medial plane is considered the plane ~1 μm below the apical surface. Scale bar in A = 10 μm for A–F. Scale bar in A' = 1 μm for A'–F'. (G, H) Manders' coefficient was calculated and plotted for individual worms in *nekl-2::mNeonGreen*; *nekl-3::mKate* strains. The fraction of NEKL-2::mNeonGreen puncta overlapping with NEKL-3::mKate puncta in the apical and medial plane (G) and vice versa (H) are shown. Raw data are available in S1 File.
(TIFF)

**S2 Fig. Colocalization of NEKLs with clathrin-coated pits.** (A–F) Colocalization assays were performed on strains expressing either NEKL-2::mNeonGreen (A–C) or NEKL-3::mNeon-Green (D–F) with APA-2::mScarlet, the alpha subunit of the AP2 adaptor complex, which is present in clathrin-coated pits. Note that these images were collected on a different confocal microscope and thus appear somewhat different than images in the paper (see Materials and Methods). Scale bar in A = 10 μm in A–F.
(TIFF)

**S3 Fig. Effects of NEKL depletion on Golgi compartments.** (A) Representative confocal images of $P_{hyp7}$::AMAN-2::mNeonGreen expression in the indicated backgrounds. (B, C) Mean intensity of $P_{hyp7}$::AMAN-2::mNeonGreen expression (B) and the mean number of $P_{hyp7}$::AMAN-2::mNeonGreen-positive vesicles (C) for individual worms were plotted in the graphs. (D) Representative confocal images of $P_{hyp7}$::AMAN-2::mNeonGreen co-expressed with a functional multi-copy NEKL-3::mCherry reporter. Error bars represent the 95% confidence intervals. p-Values were obtained by comparing means using an unpaired t-test: ****$p < 0.0001$, ***$p < 0.001$; ns, not significant ($p > 0.05$). Raw data are available in S1 File.
(TIFF)

**S4 Fig. Three-dimensional rendering of effects of NEKL deletion on basolateral cargoes.** (A–C) Three-dimensional plots showing the individual pixel intensity along the *x* and *y* planes for the fluorescent images presented in Fig 3C–3E. (D,E) Three-dimensional plots showing the pixel intensity value along the *x* and *y* planes for the fluorescent images presented in Fig 3F and 3G. We note that the basal boundary occurs in the *x* axes at approximately 40 μm and 10 μm whereas the seam cell boundary occurs between 20–30 μm.
(TIFF)

**S5 Fig. Effects of cup-5 loss of function on a membrane cargo, TGN-38.** Representative images of $P_{hyp-7}$::TGN-38::GFP expression in auxin-treated day-2 adults in wildtype and *cup-5* mutant background. Red arrows indicate accumulation of cargoes in vesicle-like structures. Mean intensity values of the $P_{hyp-7}$::TGN-38::GFP expression for individual worms are plotted in the graph. Error bars represent the 95% confidence intervals. p-Values were obtained by comparing means using an unpaired t-test: ****$p < 0.0001$, ns, not significant ($p > 0.05$). Raw data are available in S1 File.
(TIFF)

**S6 Fig. Effects of NEKL depletion on cargo trafficking near the neural membrane.** Representative images of $P_{hyp-7}$::MIG-14::GFP expression in auxin-treated wild-type, *nekl-2::aid*, and *nekl-3::aid* day-2 adults in the apical/medial plane. Red arrows show the presence of MIG-14::GFP expression in wild-type and *nekl-2::aid* adults at the hyp7 membrane surrounding the ALM neuron. The red arrowhead indicates the absence of MIG-14::GFP expression in *nekl-3:: aid* worms. Scale bar = 10 μm in the three lower-magnification images. The yellow

bracket along the ALM neuron in wild type indicates the region show in the higher magnification image (yellow outline). The bar graph shows the percentage of worms exhibiting $P_{hyp-7}$:: MIG-14::GFP expression surrounding the ALM/PLM neurons in the indicated backgrounds. Numbers of worms are indicated for each bar. Error bars represent the 95% confidence interval. p-Values were obtained using Fisher's exact test: ****p < 0.0001. Raw data are available in S1 File.
(TIFF)

**S7 Fig. Effect of NEK6 or NEK7 depletion in early endosomal compartments in glioblastoma cells.** Mock-transfected cells, NEK6 siRNA-transfected cells, or NEK7 siRNA-transfected cells were plated on coverslips and immunostained with antibodies against the sorting endosome marker protein EEA1. Average sizes of early/sorting endosomes are shown for the three conditions, with significant differences observed between the mock-transfected cells and both siRNA treatments as well as between NEK6 siRNA and NEK7 siRNA-transfected cells. Yellow box in panels indicates region of enlarged inset. Scale bars = 10 μm.
(TIFF)

**S8 Fig. Effects of NEK6 or NEK7 depletion on recycling endosomes in glioblastoma cells.** Glioblastoma cells were mock transfected or were transfected with NEK6 siRNA or NEK7 siRNA. Transfected cells were then plated on coverslips and immunostained with antibodies against the tubular recycling endosome marker protein MICAL-L1. Average numbers of MICAL-L1 tubular endosomes per field were determined for the three conditions, with significant differences observed between the mock-transfected cells and both siRNA treatments. Yellow box in panels indicates region of enlarged inset. Scale bars = 10 μm.
(TIFF)

**S1 Table. List of all the strains used in this study.**
(PDF)

**S1 File. Compilation of raw data used in this study.**
(XLSX)

## Acknowledgments

We thank Amy Fluet for editing this manuscript.

## Author Contributions

**Conceptualization:** Braveen B. Joseph, Shaonil Binti, Barth D. Grant, Steve Caplan, David S. Fay.

**Data curation:** Braveen B. Joseph, Naava Naslavsky, Shaonil Binti, Sylvia Conquest, Lexi Robison, Rafael O. Homer, Steve Caplan, David S. Fay.

**Formal analysis:** Braveen B. Joseph, Naava Naslavsky, Shaonil Binti, Sylvia Conquest, Lexi Robison, Rafael O. Homer, Steve Caplan, David S. Fay.

**Funding acquisition:** Barth D. Grant, Steve Caplan.

**Investigation:** Braveen B. Joseph, Naava Naslavsky, Shaonil Binti, Sylvia Conquest, Ge Bai, Rafael O. Homer, David S. Fay.

**Methodology:** Braveen B. Joseph, Naava Naslavsky, Shaonil Binti, Sylvia Conquest, Lexi Robison, Ge Bai, Rafael O. Homer, David S. Fay.

**Project administration:** Barth D. Grant, Steve Caplan, David S. Fay.

**Resources:** Barth D. Grant, Steve Caplan, David S. Fay.

**Supervision:** Braveen B. Joseph, Barth D. Grant, Steve Caplan, David S. Fay.

**Validation:** Braveen B. Joseph, Naava Naslavsky, Shaonil Binti, Sylvia Conquest, Lexi Robison, Ge Bai, Rafael O. Homer, David S. Fay.

**Visualization:** Braveen B. Joseph, Naava Naslavsky, Shaonil Binti, Sylvia Conquest, Lexi Robison, Rafael O. Homer, David S. Fay.

**Writing – original draft:** Braveen B. Joseph, Shaonil Binti, Steve Caplan, David S. Fay.

**Writing – review & editing:** Shaonil Binti, Barth D. Grant, Steve Caplan, David S. Fay.

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
