## [Decision Letter · Decision Letter 0]

6 Feb 2023

Dear Dr Fay,

Thank you very much for submitting your Research Article entitled 'Conserved NIMA kinases regulate multiple steps of endocytic trafficking' to PLOS Genetics.

The manuscript was fully evaluated at the editorial level and by independent peer reviewers. The reviewers appreciated the attention to an important topic but identified some concerns that we ask you address in a revised manuscript.

We therefore ask you to modify the manuscript according to the review recommendations. Your revisions should address the specific points made by each reviewer. Some of the reviewers' comments may be addressed by revising the presentation, however you may feel that some points merit additional experimental work.

Yours sincerely,

Andrew D. Chisholm

Academic Editor

PLOS Genetics

Gregory P. Copenhaver

Editor-in-Chief

PLOS Genetics

Reviewer's Responses to Questions

**Comments to the Authors:**

Reviewer #1: Summary

The Fay lab previously demonstrated that the NIMA like kinases NEKL-2 and NEKL-3 localize to foci and regulate clathrin-mediated LRP-1 endocytosis at the apical membrane of the Hyp7 syncytium. Here Joseph et al further characterize the role NEKL-2 and NEKL-3 in endosome trafficking. They demonstrate that NEKL-3 colocalizes with RAB-5 and RAB-7 positive endosomes while NEKL-2 shows some colocalization with RAB-5 and much less with RAB-7. NEKL-2 and NEKL-3 do not grossly colocalize indicating that they are mostly on distinct compartments. Auxin mediated depletion of NEKL-2 resulted in tubulation of RAB-5 compartments and depletion of NEKL-3 resulted in tubulation of RAB-11 compartments. The basal recycling cargo SMA-6 (clathrin dependent) and DAF-4 (clathrin independent) accumulate at or near the plasma membrane in NEKL-2 and NEKL-3 depleted animals consistent with their role in trafficking. However, intracellular cargo receptors MIG-14 and TGN-38 which originate from the Golgi are reduced in NEKL-2 and NEKL-3 depleted animals to different degrees suggesting they are being degraded in the lysosome. Consistent with that hypothesis TGN-38 levels are increased in NEKL-2 and NEKL-3 depleted animals when endosome to lysosome trafficking is impeded in a cup-5 mutant. Consistent with the NEKL-2 and NEKL-3 results, siRNA depletion of NEK6 and NEK7 (human NEKL-3 homologs) in HeLa cells results in decreased levels of Mannose-6-phosphate receptor suggesting that it too might be targeted to the lysosome. siRNA of NEK6 and NEK7 increased EEA1 early endosome size in HeLa and cells and siNEK7, but not siNEK6 does the same in Glioblastoma cells. siNEK6 and siNEK increase the area (length?) of MICAL-L1 tubules in both HeLa and glioblastoma cells. These data suggest that there is a conserved role for NIMA like kinases in endosome fission or fusion.

Strengths

This paper is well written and easy to read without notable typographical errors.

Most of the data is quantified and clearly displayed. All the image quantification data is available in an excel file.

The novelty of the study is that NIMA like kinases regulate endosome tubulation.

Experimentation in both C. elegans and human cell lines broadens the impact of the findings.

The experiments are well done.

Weaknesses

The study is a bit of a look and see approach rather than hypothesis driven. I find the tubulation phenotypes are the most dramatic and interesting as they point toward a specific function for the NEKL-2 and 3. I would have liked to see more experiments focused on the tubulation defects. A specific role for the NIMA kinases is not determined.

Supplementary data on glioblastoma cells is not quantified.

Points of consideration

NEKL-3 appears to have a reticulate localization pattern with bright foci that colocalize with endosomes. Does NEKL-3 localize to the ER? The ER contacts many membrane-bound organelles and is implicated in fission of these compartments. The roles of NEKL-2 and NEKL-3 in membrane scission suggests that they could localize to ER::endosome contacts to regulate endosome fission. This would be an interesting avenue to pursue in the future.

Mutation of cup-5 increases TGN-38 levels in NEKL-2/3 depleted animals. Does it also increase TGN-38 levels in a wild-type background? In other words, is it specific to the NEKL depleted animals?

Is kinase activity required for NEKL-2 and 3 function?

Figure 3. Is it possible to discern the basolateral membrane in wild-type animals as a point of reference using arrows? There is clearly an accumulation of SMA-6 and DAF-4, however the boundary is not clear in wild-type.

Reviewer #2: This manuscript submitted to PLoS Genetics by Joseph et al. reports the functions of two conserved NIMA-related kinases, NEKL-2 (NEK8/9 homolog) and NEKL-3 (NEK6/7 homolog), in endocytic trafficking in both C. elegans and mammalian tissue culture cells. The Fay Lab previously reported that NEKL-2 and NEKL-3 are essential for C. elegans molting and that they regulate apical clathrin-mediated endocytosis in the epidermis. In this study, the authors found that NEKL-2 and NEKL-3 are differentially localized to, and regulate the morphology of, distinct endosomal compartments. They also provided evidence that NEKL-2 and NEKL-3 control the sorting of cargoes between different intracellular compartments, and the endocytosis of two BMP receptors from the basolateral surface. Furthermore, they showed that knocking down the NEKL-3 homologs in human cell lines also affects endosomal morphology, suggesting that the functions of these NIMA related kinases in regulating endocytic trafficking is conserved.

The manuscript is very well written. It has identified new roles for members of the NIMA family of kinases, which are important in human development and disease. I have one main concern. The authors showed that depletion of NEKL-2 or NEKL-3 affects the uptake of both SMA-6, a clathrin-dependent cargo, and DAF-4, a clathrin-independent cargo, from the basolateral surface of epidermal cells. The authors speculated that these two receptors may be trapped in early endosomes near the basolateral surface. In light of their previous findings that NEKL-2 and NEKL-3 regulate apical clathrin-mediated endocytosis in the epidermis, it will be important to determine whether additional clathrin-independent cargoes are also affected upon NEKL-2 or NEKL-3 depletion. Furthermore, the data (at least those presented in Figure 2) do not seem to show accumulation of early endosomes upon NEKL-2 or NEKL-3 depletion. Maybe the accumulation of SMA-6 and DAF-4 is not due to defects in intracellular trafficking upon NEKL-2 and NEKL-3 depletion, rather, due to some other un-identified role(s) of NEKL-2 and NEKL-3 on the cell surface.

I have several additional comments listed below:

1) I wonder if Figure 1 is somehow mis-labeled. The images and the quantifications do not seem to match. The authors concluded, based on the quantification shown in panel M, that NEKL-2 has substantial co-localization with the early endosome marker RAB-5, but not with the late endosome marker RAB-7, while NEKL-3 has substantial co-localization with both RAB-5 and RAB-7. However, based on the images shown in Figure 1, NEKL-2 (but not NEKL-3) seems to have substantial co-localization with both RAB-5 and RAB-7 (1I’ and 1L’), while NEKL-3 appears to primarily co-localize with RAB-7 (1C’ and 1F’).

2) The NEKL-2::mNeonGreen and NEKL-3::mNeonGreen signals shown in Figure S1 appear to be significantly brighter than those shown in Figure S2. Is this due to differences in different focal planes imaged or to other reasons?

3) In Figure 3, the diameters of NEKL-3::AID worms (panels E and G) seem to be much smaller than those of WT or NEKL-2::AID worms (panels C, D and F). Is this difference due to differences in focal planes and/or stages of the animals examined, or due to NEKL-3::AID worms being unhealthy after NEKL-3 depletion?

4) Based on the western blots shown for both NEK6 and NEK7 in Figures 5-7, there seems to be substantial variation in the efficiency of the siRNA knockdown experiments. Is there any correlation between the knockdown efficiency and the severity of the phenotypes observed? Along the same line, the antibodies for NEK7 used for the western blots shown in Figures 5-7 seem questionable. It is unclear which band is specifically NEK7. In Figure 5, NEK7 appears to be the bottom one of two bands, while in Figure 6, it appears to be the middle one of three. Surprisingly, the intensities of the bottom two bands are both drastically reduced in Figure 7. The authors may want to either repeat the western blots, or comment on the specificity of the antibody.

**Have all data underlying the figures and results presented in the manuscript been provided?**

Reviewer #1: Yes

Reviewer #2: Yes

PLOS authors have the option to publish the peer review history of their article (what does this mean?). If published, this will include your full peer review and any attached files.

Reviewer #1: No

Reviewer #2: No

---

## [Editor Report · Decision Letter 1]

11 Apr 2023

Dear Dr Fay,

We are pleased to inform you that your manuscript entitled "Conserved NIMA kinases regulate multiple steps of endocytic trafficking" has been editorially accepted for publication in PLOS Genetics. Congratulations! 

Yours sincerely,

Andrew D. Chisholm

Academic Editor

PLOS Genetics

Gregory P. Copenhaver

Editor-in-Chief

PLOS Genetics

Comments from the reviewers (if applicable):

**Data Deposition**

http://datadryad.org/submit?journalID=pgenetics&manu=PGENETICS-D-22-01467R1

**Press Queries**

---

## [Editor Report · Acceptance letter]

23 Apr 2023

PGENETICS-D-22-01467R1 

Conserved NIMA kinases regulate multiple steps of endocytic trafficking 

Dear Dr Fay, 

We are pleased to inform you that your manuscript entitled "Conserved NIMA kinases regulate multiple steps of endocytic trafficking" has been formally accepted for publication in PLOS Genetics! Your manuscript is now with our production department and you will be notified of the publication date in due course.

With kind regards,

Timea Kemeri-Szekernyes

PLOS Genetics

On behalf of:
